# Combined crystallographic fragment screening and deep mutational scanning enable discovery of Zika virus NS2B-NS3 protease inhibitors

The Zika viral protease NS2B-NS3 is essential for the cleavage of viral polyprotein precursor into individual structural and non-structural (NS) proteins and is therefore an attractive drug target. Generation of a robust crystal system of co-expressed NS2B-NS3 protease has enabled us to perform a crystallographic fragment screening campaign with 1076 fragments. 46 fragments with diverse scaffolds are identified to bind in the active site of the protease, with another 6 fragments observed in a potential allosteric site. To identify binding sites that are intolerant to mutation and thus suppress the outgrowth of viruses resistant to inhibitors developed from bound fragments, we perform deep mutational scanning of the NS2B-NS3 protease. Merging fragment hits yields an extensive set of 'mergers', defined as synthetically accessible compounds that recapitulate constellations of observed fragment-protein interactions. In addition, the highly sociable fragment hits enable rapid exploration of chemical space via algorithmic calculation and thus yield diverse possible starting points. In this work, we maximally explore the binding opportunities to NS2B-NS3 protease, facilitating its resistance-resilient antiviral development.

Zika virus (ZIKV) belongs to the *Flaviviridae* family and is closely related to other flaviviruses such as Dengue virus (DENV), West Nile virus (WNV) and Yellow Fever virus (YEV). Although ZIKV infections typically manifest with mild symptoms, outbreaks in Brazil and French Polynesia revealed that the virus can cause microcephaly in newborns and Guillain-Barré syndrome in adults[1–3]. While this development sparked significant global health concern, no vaccine or antiviral therapeutic is available for the prevention or treatment of ZIKV infection[4].

The single-stranded, positive-sense ZIKV RNA is translated to produce a single polyprotein precursor that is cleaved by host proteases and viral protease NS2B-NS3 into individual functional proteins. NS3 contains a protease active site, with a serine, histidine and aspartate catalytic triad. Approximately 40 amino acids (aa) of NS2B

function as a cofactor of NS3. These residues wrap around NS3, with the C-terminal residues forming a β-hairpin that structurally contributes to the protease active site as well as participates in substrate recognition[5]. Structural studies have revealed several conformations of this two-component NS2B-NS3 protease, specifically a closed conformation[6], an open conformation[7,8], and a super-open conformation[9], depending on the dynamic interactions between NS2B and NS3. In the open and super-open conformations, the C-terminal portion of NS2B is disordered and distant from the NS3 catalytic site, resulting in an unstructured substrate binding site. These two conformations therefore, have been considered as inactive forms, while the closed conformation of NS2B-NS3 with a well-structured binding pocket represents a promising targeting opportunity for antiviral development.

✉ e-mail: xiaomin.ni@cmd.ox.ac.uk; matthew.evans@mssm.edu; frank.vondelft@cmd.ox.ac.uk

Two strategies have been deployed to target the NS2B-NS3 protease. The first involves the development of specific inhibitors that compete with the substrate on the active site. The second strategy focuses on non-competitive inhibitors, which target allosteric sites. Despite these efforts, neither competitive nor non-competitive inhibitors have advanced to clinical trials. Most of the reported competitive NS2B-NS3 inhibitors are limited to peptide mimics, such as macrocyclic inhibitors[10,11]. Although these peptidomimetic inhibitors display nanomolar potency against ZIKV NS2B-NS3 protease, and some additionally have broad-spectrum antiviral activity, inhibiting growth of Dengue and West Nile viruses as well as Zika virus[12], their cellular activity and pharmacokinetic properties are poor. In parallel, great effort has focused on developing allosteric inhibitors that block the interactions between NS2B and NS3 and lock the protease in the inactive form[13,14]. This strategy has yielded several potent non-competitive inhibitors, including NSC135618[15], which shows antiviral activity in cellular assays. Nevertheless, the allosteric site as well as the allosteric inhibitor binding mode have not yet been properly validated due to a lack of reliable structural information.

Viruses can evolve resistance to any inhibitor if the amino acids that such an inhibitor binds exhibit sufficient tolerance to mutation, while the mutations that abrogate inhibitor binding do not greatly impair viral fitness. To prioritize NS2B-NS3-binding fragments that exhibit genetic constraint and thus are less likely to be able to evolve resistance, we performed a deep mutational scan (DMS) of the NS2B-NS3 protease. DMS is a high-throughput experimental technique that relies on deep sequencing technology to measure the effect of each possible amino acid mutation at each position of a protein. DMS has emerged as a powerful tool in virology, particularly in studies of viral fitness and evolution[16]. Recent studies of the mutational potential of viral proteins, such as the SARS-CoV-2 spike protein[17] and the ZIKV envelope protein[18], provide rich information about the sequence-function relationships that apply to antibody engineering as well as vaccine design.

In this work, we combined crystallographic fragment screening with DMS to accelerate the development of resistance-resilient antiviral therapeutics targeting ZIKV NS2B-NS3 protease. Our crystallographic fragment screening against 1076 fragments identified diverse chemical scaffolds at both the active site and a potential allosteric site, providing structural details of fragment engagement for rapid follow-up compound design. The DMS results reveal the mutational potential of the hotspots for ligand engagement, which will guide rational compound design to suppress drug resistance at early stages of medicinal chemistry.

## Results

### Establishing a suitable crystal system for fragment screening

Several constructs have been reported to efficiently generate the ZIKV NS3 protease bound to the NS2B co-factor region. These approaches include the insertion of a glycine linker between the NS2B and NS3, as shown in the construct named gZiPro (Fig. 1a), and the co-expression of the NS2B co-factor and NS3 protease in a bicistronic vector, as demonstrated in the construct named bZiPro[6] (Fig. 1a). Construct gZiPro has been commonly used for ZIKV NS2B-NS3 protease studies. However, the artificial glycine-rich linker raises concerns of altering the substrate-binding behaviour[19]. Therefore, the unlinked binary NS2B-NS3 protease is preferred for studies. In addition, bZiPro succeeded in producing a closed conformation of ZIKV NS2B-NS3 structure (PDB ID: 5GPI)[6].

To produce crystals of ZIKV NS2B-NS3 protease in closed conformation for fragment screening, we generated a modified version of the PDB structure 5GPI[6]. The model of 5GPI contains 4 protein molecules in the asymmetric unit, with the N-terminal residues of NS3 in one molecule binding to a neighbouring active site, making this unsuitable for crystal soaking. A further issue with this system was the low expression yields of construct bZiPro in our hands. To address these issues, we generated a construct, designated cZiPro (Fig. 1a). In this construct, we removed the disordered C-terminal residues of NS2B

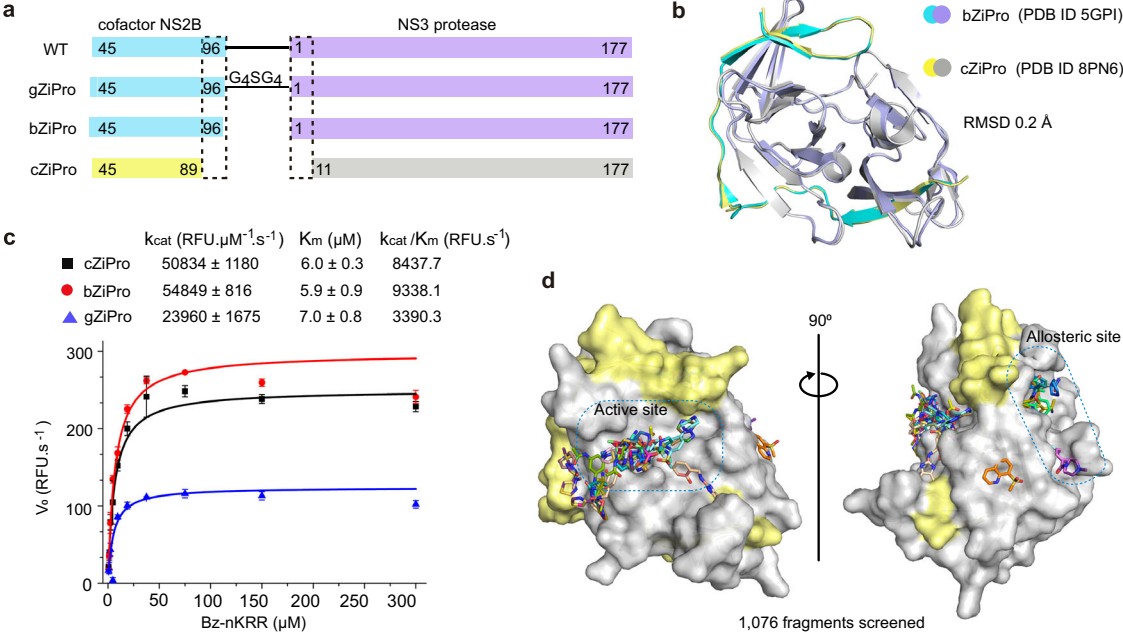

**Fig. 1 | Crystallographic fragment screening on co-expressed ZIKV NS2B-NS3 protease. a** Domain boundaries of constructs gZiPro, bZiPro and cZiPro. **b** Structure superimposition of bZiPro structure 5GPI (NS2B coloured in cyan, NS3 coloured in purple) and cZiPro structure 8PN6 (NS2B coloured in yellow, NS3 coloured in grey) with an RMSD of 0.2 Å. **c** Enzymatic activities of protease constructs cZiPro, bZiPro and gZiPro. Activities were measured using a fluorescence-dose-response assay with Bz-nKRR-AMC as substrate. Data points represent mean values, and error bars indicate standard deviations (SD) from three independent replicates (*n* = 3). **d** Surface view of ZIKV NS2B-NS3 fragment screening output in two orientations. The NS3 protein surface is coloured in grey and NS2B is coloured in yellow. Identified fragments are shown as sticks. Source data are provided as a Source Data file.

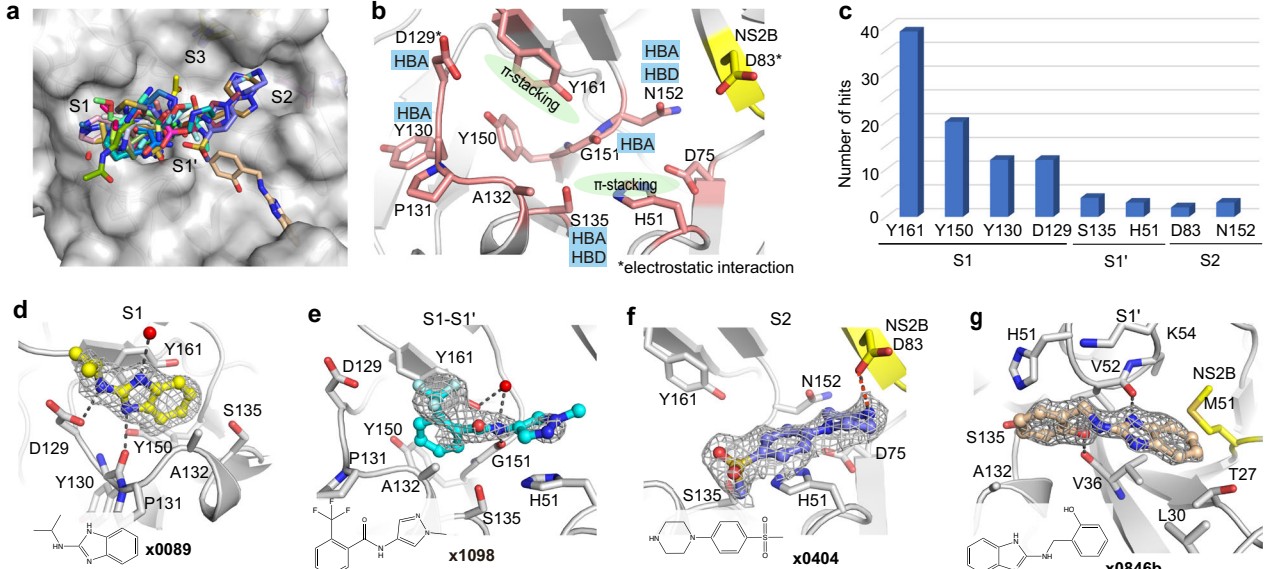

**Fig. 2 | Interaction motifs for ligand engagement in the active site of ZIKV NS2B-NS3 protease. a** Surface view of observed fragments bound in the active site. The active site is divided into subsites S1, S1', S2 and S3 based on Schechter–Berger nomenclature. **b** Key interacting residues revealed from fragment screening. Key interacting residues from NS3 are coloured in salmon, residue from NS2B is coloured in yellow. HBA: hydrogen bond acceptor. HBD: hydrogen bond donor. **c** A plot of key residues observed for fragment interaction. Y-axis represents the number of fragments forming interaction with associated residues. Residue D83 is from NS2B. Representative examples of binding pose **x0089** (**d**), **x1098** (**e**) and **x0404** (**f**) interacting at the S1, S1-S1', and S2 site, respectively. **g** The unique pose **x0846b** observed in S1' site. The PanDDA event map is shown as a dark grey mesh at 1.5 σ. Hydrogen bonds are shown as black dashed lines. Electrostatic interaction is shown in an orange dashed line. Crystal ID in bold font to represent the fragment and its binding pose. Source data are provided as a Source Data file.

and the N-terminal residues of NS3 in the 5GPI structure. This optimized construct yielded high quantities of ZIKV NS2B-NS3 protease and reproducibly produced crystals diffracting around 1.6 Å.

Our NS2B-NS3 structure (PDB code: 8PN6) was determined in $P4_322$ space group with only one protein molecule in the asymmetric unit, which is distinct from 5GPI model, yet the superposition of both structures revealed a backbone root-mean-square deviation (RMSD) of around 0.2 Å (Fig. 1b). The active site in our structure is not occluded by crystal packing, and is accessible via crystal soaking, thus making it suitable for fragment screening.

The structural similarity between cZiPro and bZiPro supports their observed functional equivalence. Both enzymes displayed comparable enzymatic activities with Km values around 6 μM and higher catalytic efficiencies than gZiPro (Fig. 1c), consistent with a previous study[19]. Moreover, all three proteases were similarly inhibited by bovine pancreatic trypsin inhibitor (BPTI), with $IC_{50}$ values near 150 nM (Supplementary Fig. 1a). bZiPro and cZiPro exhibited similar thermal stability, with melting temperatures (Tm) of 47.3 °C and 46.6 °C, respectively, as measured by Differential Scanning Fluorimetry (DSF). In contrast, gZiPro was the most thermally stable, with a Tm of 50.8 °C, likely because of its covalent linkage between the NS3 and NS2B cofactor (Supplementary Fig. 1b).

### Fragment hit identification by crystallographic fragment screening

We used our established crystal system for a crystallographic fragment screening campaign against 1076 fragments using the XChem facility at Diamond Light Source. A total of 1058 datasets with resolution ranging from 1.4–2.3 Å were collected. 51 fragments were identified from these screening efforts (see Supplementary Data 1), with 46 fragments bound in the active site and 6 in a potential allosteric site (one fragment bound to both the active site and allosteric site). Diverse chemical scaffolds were identified, including benzothiazole/benzimidazole, piperazine and quinoline-based fragments (Supplementary Fig. 2). Other moieties such as amides,

pyrazoles, and sulfones were also commonly observed among the screened fragments.

### Interaction motifs for ligand engagement in the active site
The active site of NS2B-NS3 protease site is divided into subsites S1, S1', S2 and S3 based on Schechter–Berger nomenclature[20] (Fig. 2a). Of the 46 fragments observed in the active site, 43 bound in the S1, 1 in the S1' and 2 in the S2, respectively, suggesting subsite S1 is a hotspot for fragment binding. Fragment poses in the S1 site, such as **x0089** (Fig. 2d), were frequently observed to form π-stacking with the side chain of Tyr161. **x0089** was further stabilised by three hydrogen bonds, interacting with the side chain of Asp129, the backbone of Tyr130, and a water molecule, respectively. Fragment **x1098** accommodated at both S1 and S1' site (S1-S1') (Fig. 2e), forming two π-π stacking interactions with the side chains of Tyr161 and His51. In addition, its amide motif formed a hydrogen bond with the backbone of Gly151 as well as a water-mediated hydrogen bond with the side chain of Tyr161.

S2 site binder piperazine fragment **x0404** formed an electrostatic interaction with Asp83 (NS2B), while its benzene ring formed a π-stacking interaction with His51 (Fig. 2f). We also observed one fragment pose **x0846b** (fragment bound to two sites, pose b close to catalytic residue His51), which accommodated in S1' site, forming two hydrogen bonds with backbones of Val36 and Val52, respectively (Fig. 2g). Its benzimidazole ring formed hydrophobic interaction with surrounding residues, such as Val36, Leu30, and Met51 (NS2B).

Analysis of fragment-protein interactions revealed that π-stacking with the side chain of Tyr161 is a key interaction in the S1 site, represented by 39 of 46 fragments observed in this site (Fig. 2c). Residues Asp129 and Tyr130 were also observed to form hydrogen bonds with nearly a quarter of identified fragments, together with Tyr161 and Tyr150 making the S1 site a hotspot for ligand engagement. Additionally, residues His51, Ser135 and Asn152 from NS3, and Asp83 from NS2B were also observed in fragment interactions (Fig. 2b, c), indicating their potential contribution to ligand engagement.

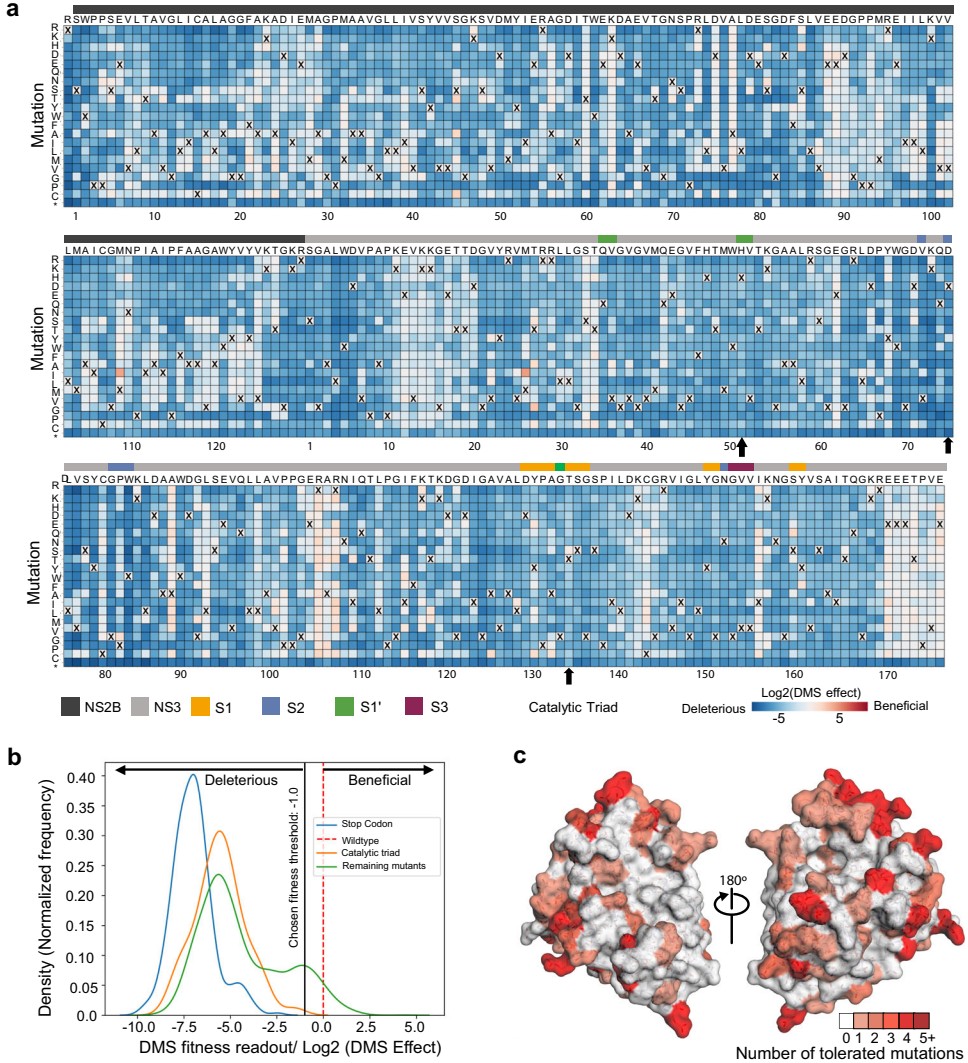

**Fig. 3 | Deep mutational scanning to measure mutational tolerance of ZIKV NS2B-NS3. a** The heatmap of NS2B-NS3 residues indicating the mutational effect of each amino acid substitutions in the ZIKV NS2B-NS3 protease. Blue mutations are deleterious for viral growth in Huh-7.5 cells relative to wildtype, white mutations are neutral, and red mutations increase growth in Huh-7.5 cells. Wildtype amino-acid identities at each site are denoted by an 'X'. In general, most mutations decrease the fitness of the virus in Huh-7.5 cells. **b** Distribution of fitness estimates for all mutations. The distributions of fitness effects of stop-codon mutants,

Catalytic triad mutants and the remaining mutants are coloured in blue, orange and green, respectively. The black vertical line indicates the threshold −1.0 (Log2 (DMS effect)) for fitness. **c** Fitness view of ZIKV NS2B-NS3 protease. Mutational tolerance is mapped onto the structure of ZIKV NS2B-NS3 (PDB ID: 8PN6). Residues marked in white do not tolerate changes to that site, while residues marked in red tolerate a range of changes. The number of changes tolerated is indicated by the number and the colour. Source data are provided as a Source Data file.

## Deep mutational scanning to measure mutational tolerance of ZIKV NS2B-NS3 protease

Given that viral proteases often evolve resistance to inhibitors through mutations in key residues, examples such as drug resistance against HIV-1 and HCV NS3/4A proteases[21–23]. Understanding the mutational tolerance of ZIKV NS2B-NS3 protease is essential for designing resistance-resilient antivirals. To gain a comprehensive view of how specific amino acid substitutions, especially at key ligand-interacting residues, affect the protease's function and ligand engagement, we employed deep mutational scanning (DMS). Specifically, we generated libraries encoding all single amino acid substitution mutants of the entire coding sequence of ZIKV NS2B-NS3 using a previously reported infectious clone of the ZIKV prototype strain MR766[24]. To ensure comprehensive coverage of all possible mutations during library cloning and screening stages, we divided the NS2B-NS3 coding sequence into three distinct 'tiles' (Supplementary Fig. 3a) and created multiple mutant plasmid libraries for each tile (three each for tiles 1

and 2, and two for tile 3) (Supplementary Data 2), which were handled separately in all subsequent steps to provide true biological replicates. Each library contained an average of $2.7 \times 10^6$ unique plasmid clones, which overrepresents the total number of possible codon variants of 3296 (32 combinations of NNK multiplied by 103 codons) by 819-fold, with an average of 1.05 mutations per clone.

Mutant virus pools were produced by transfection of HEK 293 T cells with the plasmid DNA libraries and selected for viral fitness by passaging through Huh-7.5 cells[25] (Supplementary Fig. 3b). We used deep sequencing to quantify the frequency of each mutation in the mutant viruses relative to the initial plasmid mutant libraries[26]. Stop codons, which are expected to be uniformly deleterious, were greatly purged, and nonsynonymous mutations, many of which will be deleterious, decreased in frequency (Supplementary Fig. 3c).

We next estimated the preference of each site in the NS2B-NS3 protease protein for each amino acid. These preferences represent enrichments of each amino acid at a site after selection for viral

growth, normalized to the abundance of the wild-type codon at each site. Although there is some noise, the preferences were strongly correlated among three replicates (Supplementary Fig. 3d). We experimentally quantified the relative fitness of a panel of individual NS2B-NS3 mutants and found these values highly correlated (Pearson correlation of 0.769) with the mutation effects measured by DMS (Supplementary Fig. 3e).

The mutational tolerance at any given codon can be determined from the effects of each individual mutation at that site on viral fitness. The across-replicate average of the effects of each mutation at each site in the NS2B-NS3 protease protein was calculated as the log-fold change in the frequency normalized to the change in frequency of a wild-type control and represented in the heatmap (Fig. 3a). At the lower end of fitness, stop-codon mutations and catalytic triad mutations are considered to be deleterious. Thus, their mutational effects have been applied as reference sets within the distribution of fitness effects of NS2B-NS3 mutations (Fig. 3b). Based on the above experimental testing of the fitness of individual mutants (Supplementary Fig. 3e), as well as our prior analysis of envelope protein mutational tolerance[18], we set a mutational effect threshold of fitness at -1.0 for mutations that are tolerated and allow the growth of virus, even though the fitness is reduced from that of wild-type (Fig. 3b).

Most mutations in NS2B-NS3 reduce the fitness of the virus in Huh-7.5 cells (Fig. 3a, blue squares), suggesting low mutational tolerance at many residues. Some residues, however, such as NS2B residues 88-94 and NS3 residues 12-20, exhibit high tolerance for mutation (white squares). Notably, mutations at these sites do not decrease viral fitness in Huh-7.5 cells, instead displaying fitness similar to that of viruses with wild-type residues (X) at these positions. Some sites, such as NS3 Arg106 (pink squares), showed both a high tolerance for changes and an increase in fitness when certain mutations were introduced. Residues in the protein that have a high tolerance for mutations are not good candidates for ligand binding sites, because mutations that disrupt ligand binding do not compromise viral fitness.

To facilitate analysis of these data, we generated a visualization scheme, termed 'Fitness View', to map the mutational tolerance of each residue to our crystal structure (Fig. 3c). Surfaces that are coloured in white indicate sites that are mutationally intolerant, while residues that are dark red represent a high tolerance for mutations. Overall, this fitness map provides a wealth of information about the sequence-function relationships in the NS2B-NS3 protease protein that can be used to guide the selection of key interacting residues for rational antiviral design.

## Fitness view reveals potential mutational effect on ligand engagement in the active site

Inspection of the fitness view in the active site reveals that S1' and S2 subsites are relatively intolerant of nonsynonymous mutations (Fig. 4a), while a few positions in the S1 subsite and at the C-terminus of NS2B show mutational tolerance, indicating a risk of potential resistance mutations. Specifically, Tyr130 in the S1 subsite can potentially mutate to multiple amino acids, such as cysteine, glutamic acid. At position 132, cysteine and proline substitutions are predicted to have similar replication efficiency to the wild-type alanine (Fig. 4b). Despite the high mutability of Tyr130 and Ala132, S1 fragment binders from our screen predominantly interact with the backbone of Tyr130 and have indirect contact with Ala132. Thus, the effect of the potential mutations of Tyr130 and Ala132 on ligand engagement is estimated to be limited. Similarly, the mutational flexibility of Gly82 and Phe84 at the C-terminus of NS2B is less of a concern. Phe84 hasn't been observed for fragment interaction, while Gly82 used its backbone for fragment interaction.

However, the potential mutation of tyrosine to phenylalanine at position 161 of NS3 raises our concern. Although Tyr161 contributes to fragment interaction mainly via π-stacking, which can be replaced by

phenylalanine, we also observed fragments, such as x0719 (Fig. 4c), that formed a direct H-bond with the hydroxyl group of Tyr161. Mutating tyrosine to phenylalanine would abolish the hydrogen bond formation. Thus, hydrogen bonding to Tyr161 is suggested to be avoided for follow-up compound design to decrease the probability of drug resistance. In contrast, key interacting residues Asp129 in the S1 subsite and Asp83 (NS2B) in the S2 subsite are mutational intolerant, suggesting opportunities for fragment growth (Fig. 4a, d).

## Fitness view invalidates the reported allosteric site as resistance-robust opportunity

Six fragment hits were observed outside of the orthosteric site, in a pocket located beneath the catalytic residues and next to the interface between the C-terminal β-hairpin of NS2B and NS3 (Fig. 5a). This potential allosteric site contains two sub-pockets, named as allosteric site AS1' and AS2' (Fig. 5b, c). AS1' has Trp69 and Trp83 as its major interacting residues. One example, fragment pose x0130, formed π-stacking with the side chain of Trp69 and was further stabilised by hydrogen bonding to the backbone of Trp83 (Fig. 5b).

AS2' is formed mainly by hydrophobic residues, including Phe116, Ile123, Ala164, and Ile165 from NS3, and Leu78 and Phe84 from NS2B. Four fragments with different scaffolds were observed in AS2' with different poses. Fragment x0806 contains a benzene ring that formed hydrophobic interactions with surrounding hydrophobic residues (Fig. 5c), while the nitrogen atom N2 from its triazole ring formed a hydrogen bond with the backbone of Gln167. Another hydrogen bond was formed between its amine group and a water molecule (Fig. 5c). In contrast, Fragment x0777 bound in AS2' exhibited a completely different binding mode. Its hydroxyl group formed two water-mediated hydrogen bonds with the side chains of Thr118 and Asp120 (Fig. 5d). The N2 atom from its thiadiazole ring formed another water-mediated hydrogen bond with Asp71, and the other nitrogen atom (N5) formed a hydrogen bond with water from the solvent. Its 5-carbon ring fit into a hydrophobic cavity that is formed by Ile123, Phe116 and Ala164. In conjunction, the fragments bound in this non-active site mapped a potential allosteric pocket next to the interface of the C-terminal β-hairpin of NS2B and NS3. This observation agrees with previous work of characterizing allosteric pockets in ZIKV NS2B-NS3 via docking and mutational studies[27,28] (Fig. 5e).

We next investigated the genetic flexibility of the key interacting residues in the allosteric site. Compared to the active site, the allosteric site has relatively higher mutability (Fig. 5f). In particular, the mutational flexibility of residues Asp71 and Thr118 raises our concern. Asp71 was observed to interact with fragments via hydrogen bonding (Fig. 5d). Its potential of mutating to serine would alter its charge as well as the length of the side chain, raising the risk of maintaining interactions with ligands. Similarly, the polar residue Thr118 has the potential of mutating to methionine, which has a longer side chain that may potentially narrow the pocket and crash with the ligand. Considering the potential mutational risk and underdefined key interacting residues in the allosteric site, targeting this allosteric site has been deprioritized in this study.

## Identified fragments are undetected in binding and activity assays

To further evaluate whether our screened fragment binders have on-scale binding affinity and inhibitory activity, we characterized the fragment binders using the GCI-based waveRAPID system to determine the binding affinity and a fluorescence-dose-response assay to determine their inhibitory activity[29]. waveRAPID is developed for compound and fragment screening and has demonstrated the capability to detect binding affinities in the low millimolar range[30]. Despite this, we were unable to obtain reliable results or a measurable signal to confidently validate our fragment binders using this binding assay (representative examples are shown in Supplementary Fig. 4a, b).

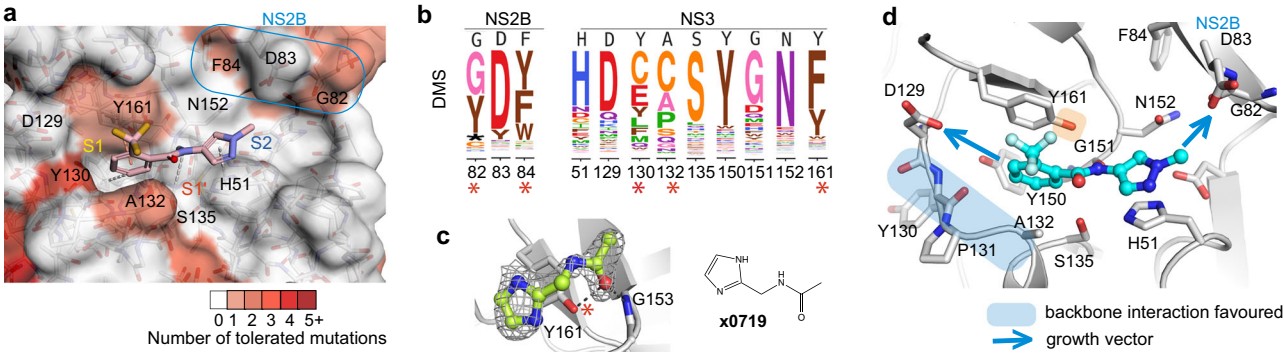

**Fig. 4 | Fitness view reveals potential mutational effect on ligand engagement in the active site. a** Fitness view of the active site with fragment **x1098** presented. Key interacting residues identified from fragment screening are numbered. The mutational intolerant region is coloured in white, and mutational tolerant region is coloured in red. **b** Logo plot of experimentally measured amino acid preferences of the labelled key interacting residues in Fig. 4a. High mutational residues are marked with an asterisk sign. **c** Example of fragment pose **x0719** formed a direct hydrogen bond with the hydroxyl group of Tyr161. **d** Vectors for fragment growth. Regions that suggest to form backbone interaction are coloured in blue. Side chain of Tyr161 is highlighted by colour orange due to its potential mutation to phenylalanine. Opportunity for fragment growth is shown as an arrow.

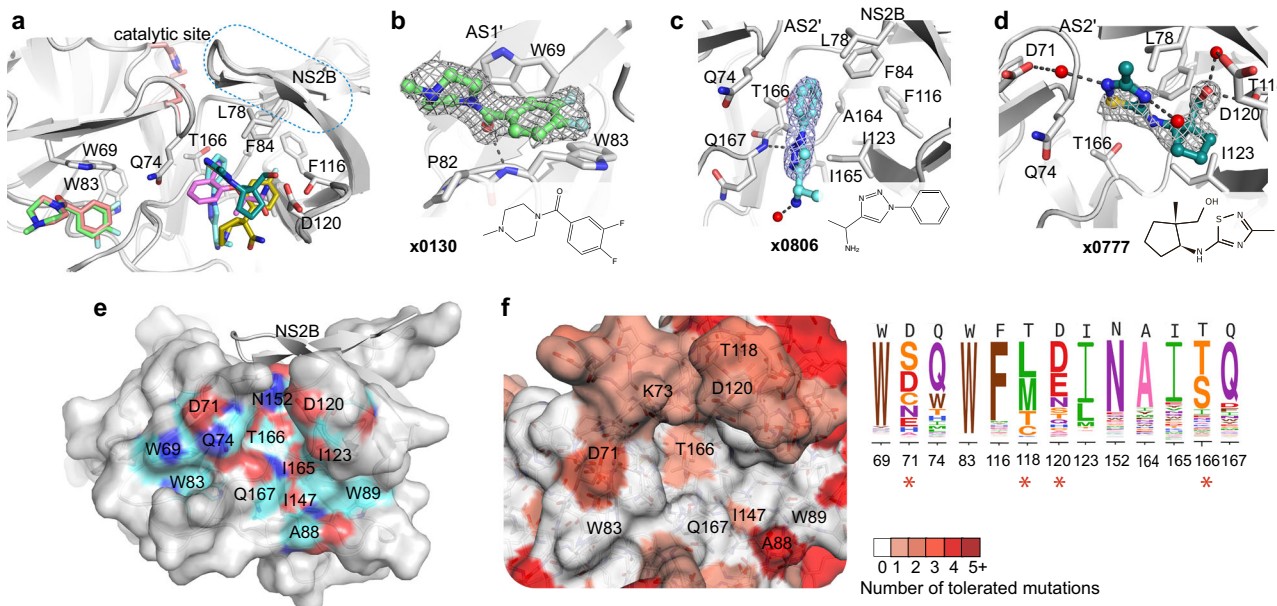

**Fig. 5 | Fitness view deprioritizes the non-active site to target. a** Overview of fragments observed in the non-active site. Fragments are shown as coloured sticks. **b** Structural details of observed pose **x0130** at allosteric site 1 (AS1'). Residues that participate in fragment interaction are shown as sticks. Representative examples of pose **x0806** (**c**) and **x0777** (**d**) bound in the AS2' site. Hydrogen bonds are shown as dashed lines. The PanDDA event map is shown as a dark grey mesh at 1.5 σ. Water molecules are shown as red spheres. **e** Surface view of an allosteric site defined by reported docking and mutational studies. Labelled residues are reported for ligand interaction. Oxygen, nitrogen and carbon atoms are coloured in red, blue and cyan, respectively. **f** Fitness view of the potential allosteric site mapping to Fig. e. The logo plot presents the experimentally measured amino acid preferences of the interacting residues in the non-active site. High mutational residues are marked with an asterisk sign.

Unsurprisingly, none of the screened fragments exhibited inhibitory activity in the biochemical assay (representative examples are shown in Supplementary Fig. 4d–f). Such observations are in common with many crystallographic fragment screens where only a low percentage of crystallographic fragment hits can be reliably detected in biophysical or biochemical assays[31]. This is further supported by a comparison study of different fragment screening assays, such as NMR, MST, and enzyme inhibitory assays, showing limited overlapping of fragment hits[32].

Notably, the positive control compound showed an $IC_{50}$ of 1.8 µM to ZIKV NS2B-NS3 in our assay (Supplementary Fig. 4g), consistent with a previous study[33]. However, its potency against DENV2 NS2B-NS3 is significantly better, with a reported $IC_{50}$ value in the low nanomolar range[34]. This observation underscores the challenge as well as the

opportunity of developing broad-spectrum inhibitors targeting flaviviral NS2B-NS3 proteases, and highlights the importance of employing parallel biochemical and biophysical assays across a panel of NS2B-NS3 proteases.

## Opportunities for rapid follow-up compound design via fragment merging and linking

Inspection of fragment-protein interaction profiling and fitness view in the active site, fragments **x0852, x1098, x0404**, and **x0846b** were selected as representative examples for merging (Fig. 6a). Fragment pose **x0852** recapitulated core interactions observed in the S1 site, specifically aromatic interaction with Tyr161, hydrogen bonding with the backbone of Tyr130. **x1098** is the only fragment observed that bridges the S1 and S2 sites with an amide motif (Fig. 6a). Its benzene

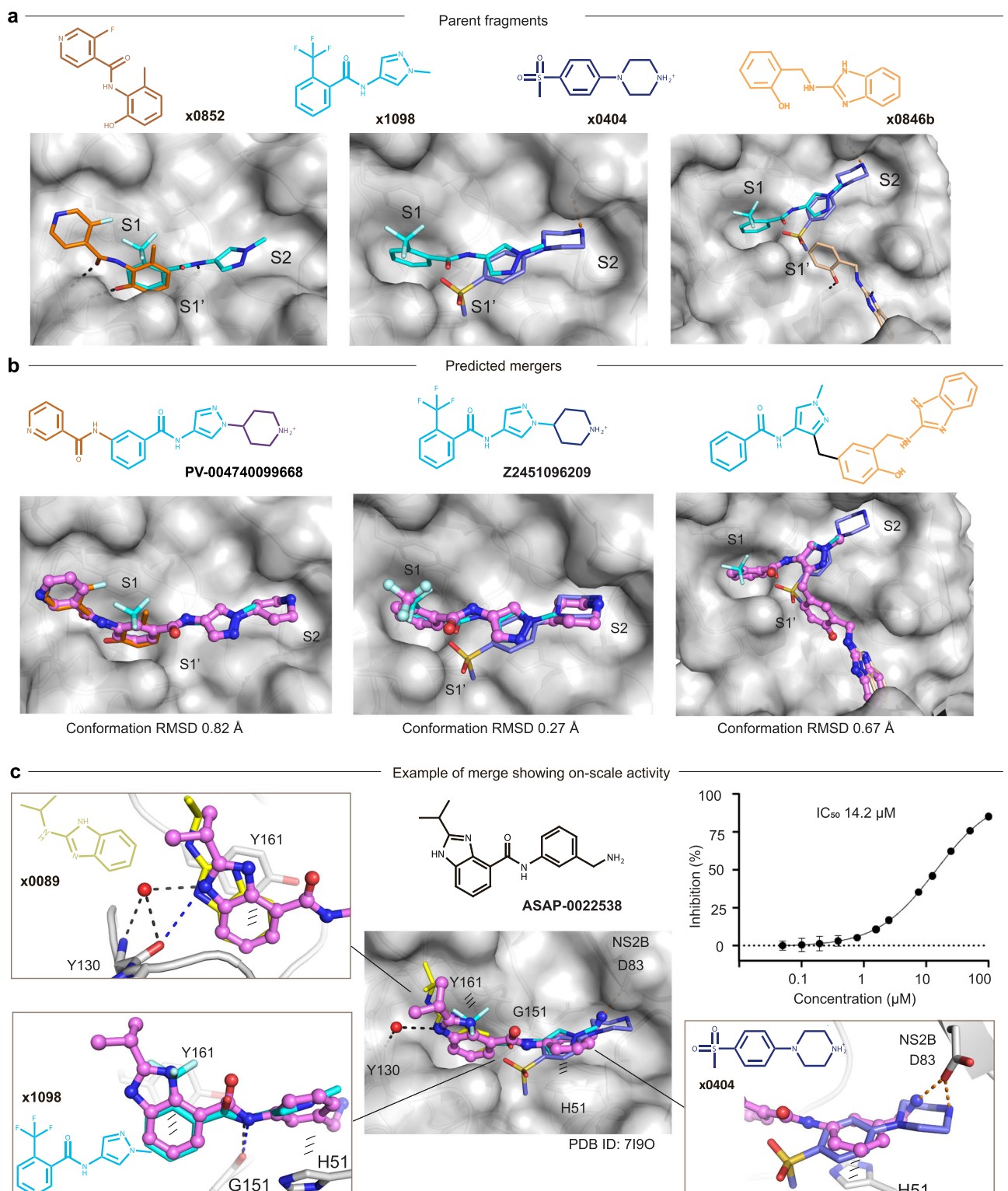

**Fig. 6 | Fragments provide merging opportunities that lead to on-scale potency. a** Fragments selected as parent hits for rapid follow-up compound design. Fragment alignment shows opportunities for merging and linking. **b** Predicted mergers posed in the active site. The 2D structure is shown at the top with its Enamine compound identity code. RMSD value indicates the conformational variance of its parent fragments. Fragments are shown in thin sticks, predicted mergers are shown in sticks and spheres. **c** A crystal structure of **ASAP-0022538** bound to ZIKV NS2B-NS3 (PDB ID: 7I9O) is shown as an example of a merge design that achieves on-scale activity (low double-digit micromolar IC$_{50}$). **ASAP-0022538**

recapitulates the poses and interactions of fragments **x0089, x1098** and **x0404**, exhibiting an IC$_{50}$ value of 14.2 μM. Water molecules are shown as red spheres. Hydrogen bonds and electrostatic interactions are shown as black and orange dashed lines, respectively. Aromatic stacking interactions are shown as titled lines. Omit map of **ASAP-0022538** is shown in Supplementary Fig.5. The IC$_{50}$ plot is shown as representative data from three independent experiments with consistent results (Source Data- Fig. 6c). Error bars in the IC$_{50}$ plot indicate standard deviations (SD) from technical duplicates in one experiment. Source data are provided as a Source Data file.

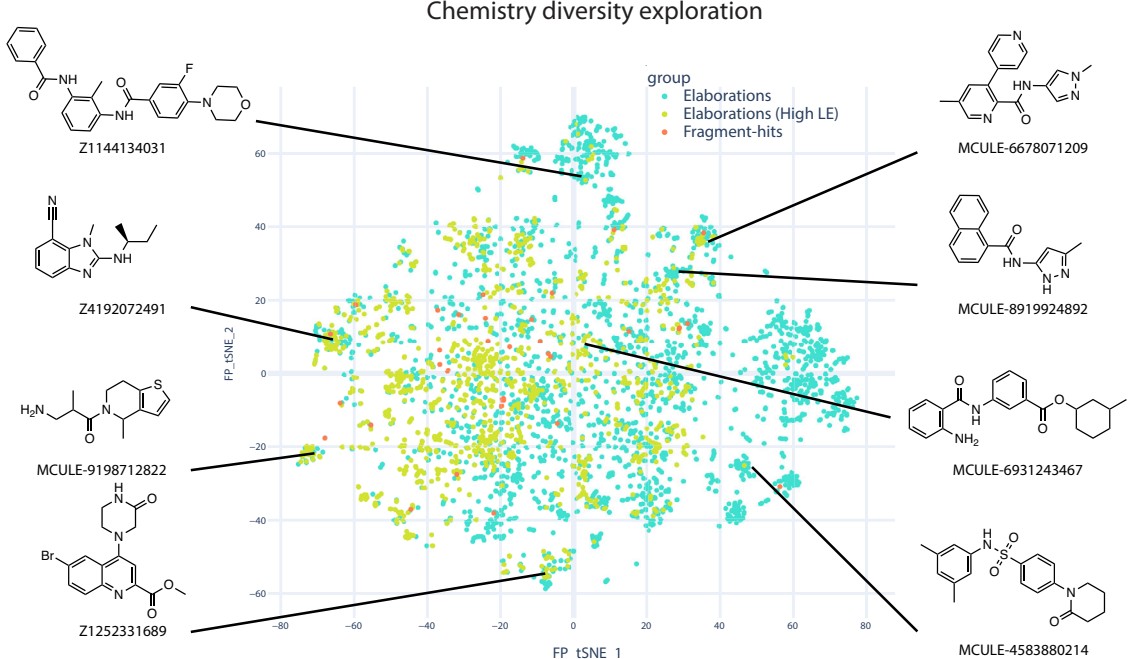

**Fig. 7 | Screened fragments are highly sociable for fragment progression.** An annotated plot of the chemical diversity of 4000 filtered catalogue compounds are close analogues (graph edit distance fewer than 6 edits) of mergers of the active site fragment-hits searched with SmallWorld in Enamine and Mcule catalogues. Source data are provided as a Source Data file.

ring nicely superimposed with the benzene ring of fragment **x0852** in the S1 site (Fig. 6a), suggesting these fragments could be merged into a single, larger scaffold. In addition, **x1098** partially overlaps with fragment **x0404** in the S2 site (Fig. 6a). Merging **x1098** and **x0404** derives a benzamide-(pyrazolyl) piperidine scaffold, which expands the small molecule to engage the S1, S1' and S2 sites simultaneously, represented by Enamine compound **Z2451096209** (Fig. 6b).

Such benzamide-(pyrazolyl) piperidine were easily accessible with over 1000 analogues. They were further assessed by an algorithmic calculation called Fragmenstein[35], which is based on the principle that the pose of mergers should preserve the poses of its parent fragments. We limited the conformational derivation of predicted mergers to be less than 1 Å in comparison with their parent hits in order to potentially maintain their parent fragments interactions. For example, the merger **PV-004740099668** has an RMSD of 0.82 Å relative to its parent fragment poses **x0852** and **x1098**, and merger **Z2451096209** has an RMSD of 0.27 Å from its parent fragment poses **x1098** and **x0404** (Fig. 6b).

To further analyse whether potency can be achieved with combined fragments, we assessed the binding pose of **ASAP-0022538** (PDB ID: 7I9O), a hit reported in a follow-up fragment-to-lead work using our screened fragments by Kenton et al.[36]. This compound recapitulated both the fragment poses and fragment-protein interactions. **ASAP-0022538** appears aligned with the poses of **x0089, x1098** and **x0404** (Fig. 6c). Its benzimidazole overlaps with fragment **x0089** in the S1 site, recapitulating the aromatic interaction and hydrogen bonding with Tyr161 and Tyr130, respectively (Fig. 6c). The amide linker of **ASAP-0022538** maintains the hydrogen bond with the backbone of Gly151 as fragment **x1098** does. The aromatic interaction between its benzene ring and His51, and the electrostatic interaction between the amine group and Asp83 (NS2B) recapitulate the interactions observed in fragment **x0404**. Combining these interacting motifs, **ASAP-0022538** was reported to show on-scale potency with an $IC_{50}$ value of 14.2 μM[36] (Fig. 6c).

Notably, the design of **ASAP-0022538** was executed separately to this work, and thus our fragment-based analysis above is retrospective. However, the utility of our fragments as the basis of inhibitor discovery was demonstrated by Kenton et al.[36], who used our fragment data to initiate and complete a full fragment-to-lead campaign. The authors reported the use of machine learning of pharmacophoric motifs combined with iterative DMTA cycles in their campaign, progressing fragments to low-nanomolar potency[36].

Manual fragment merging is relatively straightforward, but it is largely limited to pure merges[37], such as **Z2451096209** in our case, that incorporate the exact substructures of parent fragments. Merging algorithms that fully exploit chemical space, integrate key fragment-protein interactions and reliably rank compounds from possible merges are promising for identifying **ASAP-0022538**-like effective merges directly from the fragment screen. Such an approach is described in Wills et al.[37], and while that requires further refinement of the methodology, compound **ASAP-0022538** demonstrates that it is feasible.

In addition to merging, we observed that fragment **x0846b**, which is uniquely accommodated in the S1' site, provides linking opportunities with **x1098** via a methyl linker without causing strain or internal clash (Fig. 6b). This is supported by the RMSD value of 0.67 Å compared to its parent fragments while keeping a reasonable angle (112°) of the methyl linker. Such a binding region has not been explored for small molecule engagement. Our observation, despite only one fragment, brings opportunities for diverse scaffold inhibitor development.

To further explore chemistry diversity, we used all screened fragments as input for merging using Fragmenstein[35]. Automated merging, catalogue searching and constrained docking revealed a large number (around 4000 filtered catalogue compounds) and diversity of possible mergers that could be explored (Fig. 7), such as benzimidazole-based compound **Z4192072491** (Enamine) and pyrazole naphthalene **MCULE-8919924892**. Both the mergers and the analogue docking were computed with Fragmenstein and the final compounds were predicted to be faithful to the parent fragments (RMSD < 1 Å) without violating the energy scoring function (ΔGcalc of binding < −5 kcal/mol). The abundant and diverse chemical matter suggested from algorithmically merging demonstrates that our screened fragments are highly sociable[38], enabling many possible avenues of exploration.

## Discussion

We obtained a robust ZIKV NS2B-NS3 crystal system that enabled a successful crystallographic fragment screen. The sampled fragment hits provide diverse possible starting points for rapid follow-up compound design. This brings opportunities to this antiviral target, where currently the orthosteric inhibitors are largely limited to peptidomimetics[39]. In addition, analysis of fragment-protein interaction profiling highlights the S1 subsite as the major targeting opportunity for orthosteric inhibitor development. This is supported by over 80% of observed fragments being accommodated in this region. Such a high rate of hits observed in the S1 site is likely due to the deep pocket mainly formed by the bulky side chains of Tyr150 and Tyr161.

Drug resistance remains one of the major challenges in developing effective protease inhibitors. Lessons from previous drug discovery campaigns, such as those targeting HIV-1 protease[40], highlight the importance of understanding how key interacting residues tolerate mutations. To address this, we applied DMS to profile the mutational landscape of our target protein. This approach aids in assessing fragment-protein interactions and prioritizing fragments for progression. Crystallographic fragment screening, due to its high sensitivity, often samples dozens of fragment binders, but not all of which can be effectively progressed to lead. Selection of fragment hits as promising chemical starting points requires careful filtration. Mutational profiling can serve as a powerful filter for fragment selection, particularly for antiviral targets. For instance, fragments like **x0719** that interact with highly mutable residues may be deprioritized for fragment progression, as such interactions can be readily compromised by viral evolution.

Moreover, the mutational profile constrains the binding region for ligand engagement, providing rational guidance for fragment growth. For example, the mutationally intolerant residues such as Asp129 in the S1 site and Asp83 (NS2B) in the S2 site suggest a growth vector for fragment elaboration. We propose that incorporating DMS in the early stage of medicinal discovery provides rational guidance for fragment progression and supports resilient antiviral compound design.

Multiple allosteric sites of Flaviviral NS2B-NS3 protease have been proposed, but without structural description[9,15,41]. Our fragment screening data presented herein help to reduce this knowledge gap and map the allosteric site with structural details. Although the allosteric binding pocket we described here presents in a closed conformation, it agrees with a proposed allosteric site reported previously[9,15], supporting the 'ligandability' of the interface of the C-terminal β-hairpin of NS2B and NS3. However, the distinct binding poses and interactions observed from our screening indicate the challenges for non-competitive inhibitor development. More importantly, DMS data revealed a high mutability of several interacting residues in this allosteric site, suggesting potential risks of drug resistance. We, therefore, prioritized the orthosteric site as the antiviral target. We hope this observation will provide an important insight into this field given that there is increasing effort in allosteric inhibitor development for Flavivirus NS2B-NS3 protease[39].

Merging fragments can effectively lead to on-scale affinity from non-measurable weak binders[42], and is therefore commonly applied for rapid follow-up compound design. This has been proven by several successful fragment-to-lead case studies, with targets including the SARS-CoV-2 main protease[29] and NSP3 macrodomain[43]. In this study, our screened crystallographic fragments show neither measurable binding affinity nor inhibitory activity, not unexpectedly. This is not surprising as crystallographic fragment screen is a highly sensitive assay, likely due to the high concentration (100 mM in crystal drops) used in soaking. However, the structural details of fragment-protein interactions provide clear routes, or rich information to grow fragments for higher affinity. In our case, a key fragment **x1098** bridges the S1 and S2 subsites and provides an essential pose for merging. Merging this fragment with S1 binders derives an easily accessible benzamide-(pyrazolyl) piperidine scaffold that offers a potential route for rapid follow-up compound design.

In conclusion, our integrative approach leveraging crystallographic fragment screening and deep mutational scanning provides both structural and functional insights into ligand interactions with ZIKV NS2B-NS3 protease. While the fragment screen mapped both orthosteric and allosteric binding pockets, the high mutability of key allosteric site residues identified through DMS underscored the risk of resistance and informed our prioritization of the orthosteric site. Although no initial fragment hits displayed measurable binding affinity or inhibition, structural elucidation revealed promising starting points for fragment merging, particularly through the S1–S2 bridging fragment **x1098**. These findings highlight not only the complexity of targeting this viral protease but also the power of structure-guided strategies in advancing rational antiviral design. Our work lays a foundation for the development of more robust and resistance-resistant flaviviral protease inhibitors.

## Methods

### Protein expression, purification and crystallization

A bicistronic construct here named cZiPro was based on the sequence of the PDB model (PDB ID:5GPI). The construct was created by using a synthetic *E. coli* codon-optimised gene ZIKV sequence as template (accession YP 009227202.1 in GenBank). Golden gate cloning was used to insert this into the pNIC-HIS6-GST-TEV-GG vector[44]. The final construct contains a GST fusion with a TEV site followed by the NS2B peptide (residues 45-89 aa). This is followed by an intervening ribosome binding site and the NS3 protease domain (residues 11-177aa). Cells cultured in TB media were initially grown at 37 °C, and they were induced with 0.5 mM IPTG at 18 °C overnight. Harvested cells were lysed by sonication in a buffer containing 50 mM HEPES, pH 7.5, 500 mM NaCl, 20 mM imidazole, 5% glycerol and 1 mM tris (2-carboxyethyl) phosphine (TCEP). The recombinant protein was initially purified by $Ni^{2+}$-affinity chromatography. The His6 and GST tag were removed by TEV protease treatment, and the cleaved protein was passed through $Ni^{2+}$ affinity beads and further purified by size exclusion chromatography using an HPLC column in the buffer containing 25 mM HEPES pH 7.5, 150 mM NaCl, 0.5 mM TCEP and 5% glycerol. The co-expressed protein NS2B-NS3 was concentrated to ~15 mg/ml for crystallisation. Crystallisation was performed using the sitting-drop vapour diffusion method at 20 °C. The crystals were obtained in the condition containing 30% w/v PEG2000, 0.2 M ammonium sulfate, 0.1 M sodium acetate, pH 4.8

### Crystallographic fragment screening, data collection and analysis

The fragment screening was performed using the XChem facility at Diamond Light Source, UK. The fragments from the DSi-Poised Library[45], MiniFrags Probing Library[46], CovHetFrags[47], and SpotXplorer[48] were dispensed into crystal drops by ECHO Liquid Handler with 20% (v/v) DMSO in the final condition and incubated for three hours at 20 °C. Crystals were harvested using Crystal Shifter[49] (Oxford Lab Technologies) and cryo-cooled in liquid nitrogen.

Diffraction data were collected at the I04-1 beamline at Diamond Light Source at 100 K and processed with automated pipelines using combined software, including XDS[50], Autoproc[51], Xia2[52] and DIALS[53]. All further analysis was performed using XChemExplorer[54]. Ligand restraints were generated with ACEDRG[55] and GRADE[56]. Fragment hit identification was analysed by PanDDA[57]. The electron density map was generated by Dimple[58]. Model building and refinement were carried out in COOT[59] and Buster[60] via the XChemExplorer platform.

### Cell lines and antibodies

HEK 293T and Huh-7.5[25] (provided by Charles M. Rice, Rockefeller University) cells were grown in Dulbecco's modified Eagle's medium (DMEM; Gibco BRL Life Technologies, Gaithersburg, MD) with 10% foetal bovine serum (FBS; Gibco BRL Life Technologies). The

humanized monoclonal antibody D1-4G2-4-15 (4G2) (Absolute Antibody, Oxford, UK) is a broadly reactive flavivirus antibody that binds to an epitope at the fusion loop domain of the E protein[61].

## Deep mutational scanning of ZIKV NS2B-NS3 protease

Here we present a compressed summary of the methods used for ZIKV NS2B-NS3 protease DMS and the validations of the results. Detailed methods can be found in the Supplementary Information file.

To generate mutant plasmid libraries, we created NS2B-NS3 protease codon-mutant DNA fragments using a previously described PCR mutagenesis approach[62] with sets of forward and reverse oligos that randomized each codon with an NNK sequence where N is any nucleotide and K is either a G or a T (Supplementary Data 2). These products were cloned into our previously described single-plasmid reverse genetics system for ZIKV strain MR766 (sequence is available at Genbank accession KX830961)[24] using techniques like those described in our previous ZIKV envelope protein DMS paper[18]. We generated infectious virus stocks of these ZIKV NS2B-NS3 protease DMS libraries by transfecting plasmids into HEK 293T cells using a protocol to maintain library complexity as previously reported[18,24,63]. Wild-type virus was rescued in parallel as a control. These stocks were titered on Huh-7.5 cells, and then selected by infecting these cells at an MOI of 0.05 infectious units per cell. Infected cells were collected at day 2 post-infection.

Total RNA was extracted from these cells, reverse transcribed, and subjected to Illumina deep sequencing using a barcoded-subamplicon sequencing approach described in previous work[64] (see also https://jbloomlab.github.io/dms_tools2/bcsubamp.html) to minimize sequencing errors. The raw deep sequencing data have been deposited in the Sequence Read Archive as BioProject PRJNA1125458. The code that performs the analyses of the deep sequencing data is available on GitHub at https://github.com/jbloomlab/ZIKV_DMS_NS3_EvansLab. This repository includes summary notebooks that provide detailed statistics like read depth and mutation frequencies for each library tile (see sub-directories of https://github.com/jbloomlab/ZIKV_DMS_NS3_EvansLab/results/summary/). Briefly, we used dms_tools2[65] (https://jbloomlab.github.io/dms_tools2/), version 2.4.14, to count the occurrences of each mutation in each sample (see https://jbloomlab.github.io/dms_tools2/bcsubamp.html for details). The amino-acid preferences were computed from these counts using the approach previously described[65] (see also https://jbloomlab.github.io/dms_tools2/prefs.html). The mutational effects are the log of the preference for the mutant amino acid divided by the preference for the wild-type amino acid. Output containing the numerical values of the counts of mutations in each sample, the amino-acid preferences, and mutational effects are processed on a per-tile basis. These data are provided in CSV file format in the GitHub repository. See the README for details on navigating the analysis output.

## GCI RAPID Kinetics for binding assay

The fragments were tested with Creoptix Wave (Malvern Panalytical) using PCH-NTA chips. First, the chip surface was conditioned with 1 M NaCl, 0.1 M borate pH 9.0 buffer for 180 s, followed by an injection of 250 mM EDTA solution for 180 s. Then the chip surface was equilibrated by injecting running buffer (10 mM HEPES pH 7.4, 150 mM NaCl, 0.05% Tween 20).

The immobilization of ZIKV NS2B-NS3 protease was performed using His capture combined with EDC/NHS conjugation. First, $NiCl_2$ (500 μM) was injected to activate NTA groups on the chip surface for 420 s. Then, the running buffer was injected twice for 60 s to wash the chip surface. EDC/NHS (Xantec) was injected to activate carboxyl groups on the chip surface and to couple the target protease. The target protease was injected at a concentration of 15 μg/ml for 420 s on the FC2, FC3, and FC4 channels. Excessively activated groups were blocked with the injection of ethanolamine (1 M pH 8.5, Xantec)

solution. The FC1 blank surface was treated the same as the FC2/FC3/FC4 active channel, except for the NS2B-NS3 protein capture step. The final NS2B-NS3 capture level was about 9000 pg/mm² on the active channels. All steps within the immobilization were performed at a 10 μl/min flow rate. A kinetic binding assay was performed through the Repeated Analyte Pulses of Increasing Duration (RAPID) kinetics assay in Creoptix. Samples were injected at 250 μM for 5 s association and 20 s dissociation at a 400 μl/min flow rate. Positive control compound was tested at 1 μM for an accurate result. DMSO correction was included in the method and injected as a 0.5% DMSO addition to the running buffer + 2% DMSO. Blank solution (the running buffer + 2% DMSO) was injected after every 5th sample. All steps were conducted at 25 °C. Data analysis was carried out with adjustment of x and y offset, DMSO calibration, and blank subtraction in WAVE control software 4. 5.18.

## Protease activity assay

The protease activity assay is carried out by monitoring the fluorescence generated upon cleavage of the fluorogenic substrate Bz-nKRR-AMC (International Peptides), which releases 7-amino-4-methylcoumarin (AMC) upon hydrolysis[66]. Reactions were performed in black 384-well Corning® microplates using a reaction buffer composed of 20 mM Tris-HCl, pH 8.5, 10% glycerol, and 0.01% Triton X-100. The protease was added at a final concentration of 5 nM to wells prefilled with reaction buffer. For kinetic analyses, substrate concentrations were varied in a serial dilution ranging from 300 μM to 0.2 μM. The kinetic parameters ($K_m$, $k_{cat}$, and $V_{max}$) were calculated by nonlinear fitting to the Michaelis–Menten equation. All experiments were performed in triplicate, and data analysis was conducted using Origin 9.0 software (OriginLab).

## Fluorescence-dose-response inhibitory activity assay

This protocol was adapted from a previous study[29] with minor modifications. Fragments were seeded into assay-ready plates (Greiner 384 low volume, cat. no. 784076) using an Echo 555 acoustic dispenser, and dimethyl sulfoxide (DMSO) was back-filled for a uniform concentration in assay plates (DMSO concentration 1.5%). Dose-response assays were performed in 12-point dilutions of twofold, starting at 300 μM. Reagents for the assay were dispensed into the assay plate in 10 μl volumes for a final volume of 20 μl.

Final reaction concentrations were 20 mM Tris pH 8.5, 0.01% Triton, 10% glycerol, 15 nM ZIKV NS2B-NS3 and 5 μM fluorogenic peptide substrate (Boc-Gly-Arg-Arg-AMC, CAS # [113866-14-1(free base)], Biosynth (FB110553)). ZIKV NS2B-NS3 protease was pre-incubated with the compounds for 2 hr followed by the addition of substrate and a further 30 mins incubation (all incubations performed at room temperature).

Protease reaction was measured in a BMG Pherastar FS with a 360/470 nm excitation/emission filter set. Raw data were mapped and normalized to high (Protease with DMSO, no compounds) and low (No Protease, no compounds) controls using Genedata Screener software. Normalized data were then uploaded to CDD Vault (Collaborative Drug Discovery). Dose-response curves were generated for $IC_{50}$ using nonlinear regression with the Levenberg–Marquardt algorithm with minimum inhibition of 0% and maximum inhibition 100%. To each run, we added the reported peptide-hybrid inhibitor compound **36** as a positive control (for more accurate $IC_{50}$ measurements of this compound, the dose response started from 100 μM).

## Reporting summary

Further information on research design is available in the Nature Portfolio Reporting Summary linked to this article.

# Data availability

Construct cZiPro has been deposited to Addgene https://www.addgene.org/228663/. cZiPro is designed based on the PDB structure 5GPI. Crystallographic coordinates and structure factors for the cZiPro

crystal structure and **ASAP-0022538**-bound structure have been deposited in the PDB with the accession codes 8PN6 and 7I9O, respectively. PDB codes of crystallographic fragment binders are listed in Supplementary Data 1. Source data are provided with this paper. The raw deep sequencing data have been deposited in the Sequence Read Archive as BioProject PRJNA1125458. Results of deep sequencing data are available on GitHub at https://github.com/jbloomlab/ZIKV_DMS_NS3_EvansLab. Source data are provided with this paper.

## Code availability

The code that performs the analyses of the deep sequencing data is available on GitHub at https://github.com/jbloomlab/ZIKV_DMS_NS3_EvansLab.

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

## Acknowledgements

Research reported in this publication was supported by the NIAID of the National Institute of Health under award number U19AI171399. The content is solely the responsibility of the authors and does not necessarily represent the official views of the National Institute of Health. Authors acknowledge Fundação de Amparo à Pesquisa do Estado de São Paulo project 2013/07600-3. Authors acknowledge Diamond Light Source for access to the I04-1 beamline and XChem facilities through proposal lb32627.

## Author contributions

X.N.: conceptualization, X-Ray data curation, formal analysis, visualization, writing—original draft, writing—review and editing; R.B.R.: DMS data curation and analysis, review and editing; A.S.G: crystallography support, writing—review and editing; M.P.F.: Fragment progressing; C.K., J.S., W.W.H., J.D.B: DMS analysis, visualization; B.H.B., P.G.M., C.W.E.T.: XChem support. M.F., S.W., E.P.W., C.G.: Protein engineering, protein production and quality control; N.L., E.C., H.B.: biochemical and biophysical assays. R.M.L., M. Winokan, W.T., A.V.C., M. Walsh: software, manuscript reviewing; M.X., scientific coordination; J.C.A., D.F., L.K., K.K.: validation, manuscript review and editing; N.T.K., A.L.: Chemistry data curation; I.D., R.S.F., G.O.: Constructs and enzymatic activity characterisation; A.v.d.: Conceptualization, manuscript review and editing; M.J.E.: conceptualization, writing—review and editing; F.v.D.: funding acquisition, conceptualization, writing—review and editing.

## Competing interests

A.S.G consults for DNDi and MMV. A.A.L. is a shareholder and employee of PostEra, which has commercial interests in the discovery, development and commercialization of therapeutics. Other authors declare no competing interests.

## Additional information

Xiaomin Ni [1,18] ✉, R. Blake Richardson[2,18], Andre Schutzer Godoy [3], Matteo P. Ferla[1], Caroline Kikawa[4,5,6], Jenke Scheen[7], William W. Hannon[8,9], Eda Capkin[10,11], Noa Lahav[12], Blake H. Balcomb[10,11], Peter G. Marples [10,11], Michael Fairhead[1], SiYi Wang[1], Eleanor P. Williams [1], Charles W. E. Tomlinson [10,11], Jasmin C. Aschenbrenner [10,11], Ryan M. Lithgo[10,11], Max Winokan[10,11], Charline Giroud[1], Isabela Dolci [3], Rafaela Sachetto Fernandes[3], Glaucius Oliva [3], Anu V. Chandran[10,11], Mary-Ann Xavier[10,11], Martin A. Walsh [10,11], Warren Thompson [10,11], Jesse D. Bloom [6,13], Nathaniel T. Kenton[14], Alpha A. Lee[14], Annette von Delft[1], Haim Barr[12], Karla Kirkegaard [15,16], Lizbé Koekemoer [1], Daren Fearon [10,11], Matthew J. Evans [2] ✉ & Frank von Delft [1,10,11,17] ✉

[1]Centre for Medicines Discovery, University of Oxford, NDM research Building, Roosevelt Dr, Headington, Oxford, UK. [2]Department of Microbiology, Icahn School of Medicine at Mount Sinai, New York, NY, USA. [3]São Carlos Institute of Physics, University of São Paulo, Av. João Dagnone, São Carlos 13563-120, Brazil. [4]Medical Scientist Training Program, University of Washington, Seattle, WA, USA. [5]Department of Genome Sciences, University of Washington, Seattle, WA, USA. [6]Division of Basic Sciences, Fred Hutch Cancer Center, Seattle, WA, USA. [7]Open Molecular Software Foundation, Davis, CA 95618, USA. [8]Division of Basic Sciences, Computational Biology Program, and Vaccine and Infectious Disease Division, Fred Hutchinson Cancer Center, Seattle, WA, USA. [9]Molecular and Cellular Biology Graduate Program, University of Washington, Seattle, WA, USA. [10]Diamond Light Source Ltd, Harwell Science and Innovation Campus, Didcot, Oxfordshire, UK. [11]Research Complex at Harwell, Harwell Science and Innovation Campus, Didcot, UK. [12]The Wohl Drug Discovery Institute of the Nancy and Stephen Grand Israel National Center for Personalized Medicine, Weizmann Institute of Science, Rehovot 7610001, Israel. [13]Howard Hughes Medical Institute, Seattle, WA, USA. [14]PostEra Inc, 1 Broadway, Cambridge, MA 02142, USA. [15]Department of Genetics, Stanford University, Palo Alto, CA, USA. [16]Department of Microbiology and Immunology, Stanford University, Palo Alto, CA, USA. [17]Department of Biochemistry, University of Johannesburg, Auckland Park, Johannesburg, South Africa. [18]These authors contributed equally: Xiaomin Ni, R. Blake Richardson. ✉e-mail: xiaomin.ni@cmd.ox.ac.uk; matthew.evans@mssm.edu; frank.vondelft@cmd.ox.ac.uk

