## [Peer Review file · Nature Communications]

Combining a crystallographic fragment screen with deep mutational scanning provides a robust basis for discovery of inhibitors of Zika virus NS2B-NS3 protease

Corresponding Author: Professor Frank von Delft

Version 0:

Reviewer comments:

Reviewer #1

(Remarks to the Author)

In this paper, the authors report the design of an unlinked Zika NS2B-NS3 protease variant, named cZipro, which provides high-resolution structures and favourable packing for soaking experiments. Successful crystallographic screening of 1,076 fragments, leading to the identification of 52 fragment-protein complexes with cZipro—47 binding to the active site and 6 to potential allosteric sites. However, none of these fragments exhibited inhibitory activity in biochemical assays. The authors provide suggestions for fragment merging and linking strategies to improve binding affinity. They also established deep mutational scanning to assess the mutational tolerance of the NS2B-NS3 protease. However, no structural or activity data are provided for these modified molecules. So, the impact of the paper is limited in its current form.

Questions and Suggestions:

1. Many of the identified fragments, particularly the 47 active-site binders, share similar chemical scaffolds. A table grouping these fragments by chemical structure would be helpful for comparison and interpretation.
2. The fragment merging and linking strategy (Figure 7) suggests promising candidates, such as Z2451096209 and PV-004740099668. Could the authors test their inhibitory activity and crystallize them with the protease? Presenting at least one confirmed complex structure would strengthen the conclusions.
3. Was the deep mutational scanning performed in the presence or absence of fragments? Would it be possible to assess mutational tolerance in the presence of selected hit fragments?
4. Is GCI RAPID Kinetics suitable for detecting weak binding interactions? If the results are not reproducible, were any optimization steps attempted? If the data remain unreliable, this method should be reconsidered in the manuscript.
5. Although it is important to see the mutational profile of ns2b-ns3 protease, however, applying this information to fragment development which is superior weak binding at this stage I believe is too early. Maybe authors can discuss on this.

Reviewer #2

(Remarks to the Author)

In this study, Ni et al. aims to develop resistance-resilient inhibitors against ZIKV NS2B-NS3 protease by performing crystallographic fragment screening and deep mutational scanning. A major innovation of this study was the generation of the cZIPro construct, which enables crystallographic fragment screening through crystal soaking. After screening 1076 fragments, the authors identified 47 fragments binding to active site and 6 to an allosteric site. Since the deep mutational scanning result showed that the key interacting residues in the allosteric site had high mutational flexibility, the authors focused on the fragments binding to the active site for merging. At the end, the authors proposed several mergers based on computational analysis. Overall, the idea of this study is quite interesting. However, my enthusiasm for this study has significantly diminished after I found that none of the mergers were experimentally validated, and that the development of resistance-resilient inhibitors was not actually achieved.

Major comments:

1. While this study used deep mutational scanning data to guide the selection of fragments, I wonder if sequence conservation analysis would provide similar guidance. For example, Tyr at position 161 could be replaced by Phe without a

fitness cost. Therefore, the authors proposed avoiding compounds that H-bond with the hydroxyl group of Tyr161. However, if Phe161 could be observed in nature, such a conclusion could also be drawn without the deep mutational scanning data.

2. The phrase “enable development of resistance-resilient inhibitors” in the title is not shown by the study. The authors did not experimentally validate any inhibitors, let alone their resistance profile.

Minor comments:

1. Page 6: references are needed for this sentence: “Given that viral proteases often evolve resistance to inhibitors through mutations in key residues”.
2. Is there a rationale of using a threshold of fitness at -1.0? For example, why not -0.5?
3. Given that the results of Figure 6A-B could not be reproduced, it will be informative to show the results of the replicate, maybe as a supplemental figure.
4. Page 9: “Ala 132” should be “Ala132”.
5. In the Supplemental file for extended methods, I believe “105” in pages 2 and 3 should be “10⁵”.

Reviewer #3

(Remarks to the Author)

This paper describes identification of a number of fragments that may serve as the basis of future inhibitors of zika virus protease. This is interesting and important work given the role of the zika virus protease in development of flavivirus-directed anti-virals. There are several very nice aspects of this paper including: 1) The new cZiPro construct. 2) The mutational scanning analysis in the intact virus is an outstanding contribution. The conclusions that the allosteric site is highly prone to resistance mutations is highly important. 3) Using a new crystallographic form that allowed identification of binding fragments at multiple sites. 4) The tool they report in figure 4 that allows analysis of the mutational sensitivity is easy to understand and useful. In spite of the strengths of this work I have several concerns that call into question the completeness of this report as outlined below:

1. The main concern about this paper is the lack of appropriate activity assays to assess the inhibition potency and binding affinity of the fragments they report. The data in Figure 6, was reported to be inconclusive and I agree. On Line 348, the manuscript mentions that they did not get conclusive and reproducible results. In my estimation, this is not an excuse for not providing the data because many other assays have been published in the literature and are routinely employed. First, measuring activity assays using a standard substrate such as Boc-GRR-AMC is extremely simple and works for even weak binders. This assay should be done for all of the binders reported in this paper. I have serious concerns about how these assays were performed. The positive control Fig 6H was reported in Reference 26 to be a 18 nM binder. In Fig 6H, it appears to be a 1.8 μ M binder (100-fold weaker than reported in the literature). This is concerning. Second, assessing at least a relative affinity is also possible for this protease. Kd my NMR chemical shift perturbations is also very simple and straightforward and does not require full assignments of the spectrum. A number of recent publications have used a very simple Sypro orange assay to at least estimate binding affinities. One of these approaches is really essential to allow readers to understand the binding affinities of these fragments.

2. This paper shows a number of linked fragments, but not a single fragment has been synthesized or tested. This gives the paper an incomplete feeling compared to other papers in this journal.

3. I have serious concerns about the nomenclature used in figure 2A. Line 161 notes reference 6 but uses standard Schecter and Berger nomenclature, which is a 50-year standard and the field, and should be appropriately reference. The colored patches in Fig 2A do not correspond to the Schecter and Berger subsites. Thus, the nomenclature is very confusing. This paper should use Schecter and Berger subsites. If they are not referring to those, then the paper should not use the standard nomenclature (S2, S3, S1') as it is misleading.

4. The activity of cZiPro relative to bZiPro and gZiPro should be reported.

Version 1:

Reviewer comments:

Reviewer #1

(Remarks to the Author)

The revised manuscript has addressed this reviewer's comments.

Reviewer #2

(Remarks to the Author)

While the authors addressed most of my previous comments, my enthusiasm for this study remains low since there is still no experimental validation of any of the predicted mergers, such as Z2451096209 and PV-004740099668.

Major comments

1. There remains a lack of an inhibitor developed using the information from the fragment screen in the present study.

Although ASAP-0022538 recapitulates some of the fragment-protein interaction, it was identified in a previous study (ref #36) rather than constructed using the fragments from the present study. The impact of this study is limited without experimental evidence that the fragment-based screen can lead to new inhibitors.

2. Relatedly, the phrase “development of Zika virus NS2B-NS3 protease inhibitors” in the title is misleading. First, there is only one compound in this study showing inhibition activity (i.e. ASAP-0022538). Second, ASAP-0022538 was identified in a previous study using machine learning (ref #36), rather than developed in this study.

Minor comments

1. Line 417: should “ASAP-0022538 like” be “ASAP-0022538-like”?

2. ASAP-0022538 is shown as an example of “active merge” in Figure 6C. However, the definition of “active merge” is unclear here.

Reviewer #3

(Remarks to the Author)

My concerns have been adequately addressed.

Response to reviewer comments

We would like to sincerely thank the reviewers for their valuable feedback and constructive comments to improve this manuscript. We carefully considered all points raised by the reviewers, and as suggested have now included a small molecule that recapitulated the poses and key interactions of screened fragments to strengthen the manuscript.

In addition to addressing the reviewers' direct comments, we also made some modifications to the figures to improve the overall clarity of the manuscript. Modifications made in the main text have been highlighted yellow.

Please find below point-by-point responses to the reviewers. We hope that the revisions meet the reviewers' expectations.

REVIEWER COMMENTS

Reviewer #1 (Remarks to the Author):

In this paper, the authors report the design of an unlinked Zika NS2B-NS3 protease variant, named cZipro, which provides high-resolution structures and favourable packing for soaking experiments. Successful crystallographic screening of 1,076 fragments, leading to the identification of 52 fragment-protein complexes with cZipro—47 binding to the active site and 6 to potential allosteric sites. However, none of these fragments exhibited inhibitory activity in biochemical assays. The authors provide suggestions for fragment merging and linking strategies to improve binding affinity. They also established deep mutational scanning to assess the mutational tolerance of the NS2B-NS3 protease. However, no structural or activity data are provided for these modified molecules. So, the impact of the paper is limited in its current form.

Questions and Suggestions:

- 1. Many of the identified fragments, particularly the 47 active-site binders, share similar chemical scaffolds. A table grouping these fragments by chemical structure would be helpful for comparison and interpretation.**

Response: We include a new Supplementary Fig. 2 which shows the shared chemical scaffolds of observed fragments as well as examples of fragment poses observed for interaction. While 2D chemical structures is helpful for interpretation, we put more weight on the 3D structural poses of fragments in our comparisons as chemical similar fragments might bind differently. Crystal ID is used to label each fragment and its binding pose in the main text.

- 2. The fragment merging and linking strategy (Figure 7) suggests promising candidates, such as Z2451096209 and PV-004740099668. could the authors test their inhibitory activity and crystallize them with the protease? Presenting at least one confirmed complex structure would strengthen the conclusions.**

Response: A new crystallographic structure (PDB ID: 7I9O) of small molecule ASAP-0022538 bound to the active site has been added to Fig. 6 in the revised manuscript. This compound recapitulated the binding poses and interactions of fragments **x0089**, **x1098** and

x0404, having an IC₅₀ value of 14.2 μM. Additionally, other suggested candidate compounds in Fig. 6, such as Z2451096209 and PV-004740099668, are easily accessible with over 1000 analogues, providing rich alternative chemical options. This further highlights the productive outcome of our fragment screen.

3. Was the deep mutational scanning performed in the presence or absence of fragments? Would it be possible to assess mutational tolerance in the presence of selected hit fragments?

Response: The deep mutational scanning (DMS) was performed in the absence of fragments. In our experimental system, mutational effects are inferred from changes in viral population size following cellular infection. Deleterious mutations reduce viral fitness and are thus depleted from the population, whereas tolerated mutations are maintained at frequencies similar to wild type. This was clarified in Page 7, Paragraph 2.

We agree that DMS in the presence of selected compounds can be a powerful tool to assess a virus's capacity to evolve resistance to inhibitors. Indeed, we previously used DMS of the ZIKV envelope protein to define all possible escape mutations to a panel of ZIKV-neutralizing antibodies (Sourisseau *et al.*, Journal of Virology, 2019). However, this type of selection requires a single, potent inhibitor that can be applied during mutant virus passaging to exert strong selective pressure. Because the fragment hits in this study exhibit no measurable inhibitory activity (as shown in Supplementary Fig. 4), they are not suitable for inclusion in DMS-based viral fitness assays. Even if a fragment had modest activity in cells, its inhibitory effects could confound interpretation by reducing viral population size in a way that mimics the effect of a deleterious mutation. Thus, it is not currently feasible to accurately assess mutational tolerance in the presence of fragment inhibitors.

4. Is GCI RAPID Kinetics suitable for detecting weak binding interactions? If the results are not reproducible, were any optimization steps attempted? If the data remain unreliable, this method should be reconsidered in the manuscript.

Response: GCI waveRAPID is capable of detecting weak binders and has been successfully applied for fragment screen studies, such as identification of fragments targeting SMYD3 (EA FitzGerald *et al.*, RSC Med. Chem., 2024). We included the following paragraph in the revised manuscript to explain why we chose this technique (Page 11, paragraph 1):

“waveRAPID is developed for compound and fragment screening and has demonstrated the capability to detect binding affinities in low millimolar range (Kartal et al., SLAS Discov, 2021).”

In our study, we explored the optimization of various parameters, including sample concentrations (ranging from 250 to 400 μM) and flow rates (200-400 μl/min) to obtain good results, but were unable to obtain reproducible data. Representative results have been added in Supplementary Fig. 4. In contrast, we obtained reliable results for the positive control compound (Supplementary Fig. 4c), demonstrating the success of experimental setup and confirming the capability of this method to detect on-scale affinity binders for this protease. The irreproducible results observed with the fragments are likely due to the fragment-protein interactions being too weak to be reliably detected. This is also supported by our biochemical assay, in which none of the fragment binders exhibited inhibitory activity (Supplementary Fig. 4d-f).

Given the irreproducible data and limited information presented in Fig. 6, we moved this section to Supplementary Fig. 4. We agree that other biophysical assays could be applied to test binding affinity. However, since the fragment binders displayed very weak binding signals in our assay and did not exhibit inhibitory activity in the biochemical assay, it suggests that further characterization using alternative methods is also unlikely to provide meaningful insights at this stage. Instead, the structural details on fragment-protein interactions are more informative for guiding fragment progression.

5. Although it is important to see the mutational profile of ns2b-ns3 protease, however, applying this information to fragment development which is superior weak binding at this stage I believe is too early. Maybe authors can discuss on this.

Response: We believe that incorporation of mutational profiling into fragment-to-lead development is both necessary and beneficial to direct and help triage design ideas, particularly for the design of effective compounds against antiviral targets. To address this point, we have included the following in the manuscript in the Discussion section (Paragraph 2, 3).

*“Drug resistance remains one of the major challenges in developing effective protease inhibitors. Lessons from previous drug discovery campaigns, such as those targeting HIV-1 protease³⁸, highlight the importance of understanding how key interacting residues tolerate mutations. To address this, we applied DMS to profile the mutational landscape of our target protein. This approach aids in assessing fragment-protein interactions and prioritizing fragments for progression. Crystallographic fragment screen, due to its high sensitivity, often samples dozens of fragment binders, but not all of which can be effectively progressed to lead. Selection of fragment hits as promising chemical starting points requires careful filtration. Mutational profile can serve as a powerful filter for fragment selection, particularly for antiviral targets. For instance, fragments like **x0719** that interact with highly mutable residues, may be deprioritized for fragment progression, as such interactions could be compromised by viral evolution.*

Moreover, the mutational profile constrains the binding region for ligand engagement, providing rational guidance for fragment growth. For example, the mutational intolerant residues such as Asp129 in S1 site and Asp83 (NS2B) in S2 site suggest growth vector for fragment elaboration. We propose that incorporating DMS in early stage of medicinal discovery provides rational guidance for fragment progression and supports resilient antiviral compound design.”

Reviewer #2 (Remarks to the Author):

In this study, Ni *et al.* aims to develop resistance-resilient inhibitors against ZIKV NS2B-NS3 protease by performing crystallographic fragment screening and deep mutational scanning. A major innovation of this study was the generation of the cZiPro construct, which enables crystallographic fragment screening through crystal soaking. After screening 1076 fragments, the authors identified 47 fragments binding to active site and 6 to an allosteric site. Since the deep mutational scanning result showed that the key interacting residues in the allosteric site had high mutational flexibility, the authors focused on the fragments binding to the active site for merging. At the end, the authors proposed several mergers based on computational analysis. Overall, the idea of this study is quite interesting. However, my enthusiasm for this study has significantly diminished after I found that none of the mergers were experimentally validated, and that the development of resistance-resilient inhibitors was not actually achieved.

Major comments:

1. While this study used deep mutational scanning data to guide the selection of fragments, I wonder if sequence conservation analysis would provide similar guidance. For example, Tyr at position 161 could be replaced by Phe without a fitness cost. Therefore, the authors proposed avoiding compounds that H-bond with the hydroxyl group of Tyr161. However, if Phe161 could be observed in nature, such a conclusion could also be drawn without the deep mutational scanning data.

Response: We agree that sequence conservation can provide useful information for inhibitor design, particularly when targeting functionally constrained, conserved viral elements. However, conservation analysis is inherently limited to the variation observed among naturally circulating strains and cannot reveal the full landscape of functionally permissible mutations. This limitation is especially acute for ZIKV, where nearly all E protein sequences are >95% identical and most sites show only one observed amino acid, with only a minority showing a second variant (Sourisseau *et al.*, *J. Virol.*, 2019, Fig. 3A–B).

By contrast, deep mutational scanning (DMS) experimentally measures the fitness effects of all possible amino acid substitutions, including those not yet sampled by natural evolution. In our previous study of the ZIKV E protein, DMS revealed broad mutational tolerance at many sites, particularly in the glycan loop, even though these sites are invariant in nature. These findings demonstrate that natural conservation may reflect limited evolutionary sampling rather than functional constraint.

In the current study, the DMS of NS2B-NS3 protease similarly enables a high-resolution view of mutational tolerance. For example, we found that Tyr161 can be substituted with phenylalanine without a loss in viral fitness—indicating that hydrogen bonding to the tyrosine hydroxyl is not essential and may not represent a resistance-robust interaction for inhibitor design. This level of resolution is not achievable by conservation analysis alone and highlights the utility of DMS for anticipating escape mutations and guiding structure-based design of resistance-resilient inhibitors.

2. The phrase “enable development of resistance-resilient inhibitors” in the title is not shown by the study. The authors did not experimentally validate any inhibitors, let alone their resistance profile.

Response: The title has been changed to ‘Combined crystallographic fragment screen and deep mutational scanning for development of Zika virus NS2B-NS3 protease inhibitors’

In addition, we added a crystal structure of small molecule ASAP-0022538 in **Fig.6** to strengthen the manuscript, which recapitulated the poses and interactions of fragments **x0089**, **x1098** and **x0404**, exhibiting on-scale potency with an IC₅₀ value of 14.2 μM.

Minor comments:

1. Page 6: references are needed for this sentence: “Given that viral proteases often evolve resistance to inhibitors through mutations in key residues”.

Response: “Given that viral proteases often evolve resistance to inhibitors through mutations in key residues” has been modified as “Given that viral proteases often evolve resistance to inhibitors through mutations in key residues, examples such as drug resistance against HIV-1 and HCV NS3/4A proteases²¹⁻²³.” in the revised manuscript.

(21) King, N. M.; Prabu-Jeyabalan, M.; Nalivaika, E. A.; Schiffer, C. A. Combating susceptibility to drug resistance: lessons from HIV-1 protease. *Chem Biol* 2004, 11 (10), 1333-1338. DOI: 10.1016/j.chembiol.2004.08.010 From NLM Medline.

(22) Romano, K. P.; Ali, A.; Royer, W. E.; Schiffer, C. A. Drug resistance against HCV NS3/4A inhibitors is defined by the balance of substrate recognition versus inhibitor binding. *Proc Natl Acad Sci U S A* 2010, 107 (49), 20986-20991. DOI: 10.1073/pnas.1006370107 From NLM Medline.

(23) Soumana, D. I.; Ali, A.; Schiffer, C. A. Structural analysis of asunaprevir resistance in HCV NS3/4A protease. *ACS Chem Biol* 2014, 9 (11), 2485-2490. DOI: 10.1021/cb5006118 From NLM Medline.

2. Is there a rationale of using a threshold of fitness at -1.0? For example, why not -0.5?

Response: The fitness threshold of -1.0 (corresponding to a relative fitness of ~0.1) was chosen based on both the bimodal distribution of mutation effects (Fig. 3b) and our validation experiments. This cutoff effectively separates highly deleterious mutations—such as those introducing stop codons or disrupting catalytic triad residues—from more tolerated substitutions.

Importantly, we previously found in our ZIKV E protein DMS (Sourisseau *et al.*, *J. Virol.*, 2019) that mutations with fitness values at or below 0.1 were consistently highly impaired or non-viable, whereas those above this threshold often maintained near-wild-type fitness. This empirical observation held true for the NS2B-NS3 protease as well, where our individual mutant validation studies showed that mutations below this threshold reliably led to strongly attenuated virus, while many mutations above it were functionally tolerated. Thus, the -1.0 threshold represents a biologically meaningful cutoff that balances sensitivity and specificity in defining mutational intolerance.

By contrast, using a more stringent threshold (e.g., -0.5) would overestimate intolerance and obscure regions of functional flexibility, while a more permissive cutoff (e.g., -1.5) might

misclassify critical interaction sites as mutable and weaken structure-guided inhibitor design efforts. The selected threshold thus provides a rational and validated means to interpret mutational constraint across the protease.

We have addressed this comment by editing the following in the revised manuscript (Page 7, line 245):

“Based on the above experimental testing of the fitness of individual mutants (Supplementary Fig. 1e), as well as our prior analysis of envelope protein mutational tolerance (Sourisseau et al., J. Virol., 2019), we set a mutational effect threshold of fitness at -1.0 for mutations that are tolerated and allow the growth of virus, even though the fitness is reduced from that of wild-type (Fig. 3b).”

3. Given that the results of Figure 6A-B could not be reproduced, it will be informative to show the results of the replicate, maybe as a supplemental figure.

Response: Given the irreproducible data and limited information presented in Figure 6, we have moved it to Supplementary Fig. 4. The repeat results have been added in Supplementary Fig. 4a, b.

4. Page 9: “Ala 132” should be “Ala132”.

Response: “Ala 132” has been corrected as “Ala132” accordingly.

5. In the Supplemental file for extended methods, I believe “105” in pages 2 and 3 should be “10⁵”.

Response: “105” has been corrected as “10⁵” in Pages 2 and 3.

Reviewer #3 (Remarks to the Author):

This paper describes identification of a number of fragments that may serve as the basis of future inhibitors of zika virus protease. This is interesting and important work given the role of the zika virus protease in development of flavivirus-directed anti-virals. There are several very nice aspects of this paper including: 1) The new cZiPro construct. 2) The mutational scanning analysis in the intact virus is an outstanding contribution. The conclusions that the allosteric site is highly prone to resistance mutations is highly important. 3) Using a new crystallographic form that allowed identification of binding fragments at multiple sites. 4) The tool they report in figure 4 that allows analysis of the mutational sensitivity is easy to understand and useful. In spite of the strengths of this work I have several concerns that call into question the completeness of this report as outlined below:

1. The main concern about this paper is the lack of appropriate activity assays to assess the inhibition potency and binding affinity of the fragments they report. The data in Figure 6, was reported to be inconclusive and I agree. On Line 348, the manuscript mentions that they did not get conclusive and reproducible results. In my estimation, this is not an excuse for not providing the data because many other assays have been published in the literature and are routinely employed. First, measuring activity assays using a standard substrate such as Boc-GRR-AMC is extremely simple and works for even weak binders. This assay should be done for all of the binders reported in this paper. I have serious concerns about how these assays were performed. The positive control Fig 6H was reported in Reference 26 to be a 18 nM binder. In Fig 6H, it appears to be a 1.8 uM binder (100-fold weaker than reported in the literature). This is concerning. Second, assessing at least a relative affinity is also possible for this protease. Kd my NMR chemical shift perturbations is also very simple and straightforward and does not require full assignments of the spectrum. A number of recent publications have used a very simple Sypro orange assay to at least estimate binding affinities. One of these approaches is really essential to allow readers to understand the binding affinities of these fragments.

Response: To address reviewer's concern regarding our activity assay, the details of the assay have been clarified in the Method section (subtitled as: Fluorescence-dose-response inhibitory activity assay). As suggested, we conducted activity assays using a standard peptide substrate Boc-GRR-AMC to test the fragment binders. The data in Fig. 6 (it has been moved to Supplementary Fig. 4 in the revised manuscript) are representative examples of our observations from the assays. We have modified Supplementary Fig. 4 and included the following in the revised manuscript to make this point clear (Page 11, paragraph 1):

“Despite this, we were unable to obtain reliable results or measurable signal to confidently validate our screened fragment binders using this binding assay (representative examples are shown in Supplementary Fig. 4a-b). Unsurprisingly, none of the screened fragments exhibited inhibitory activity in the biochemical assay (representative examples are shown in Supplementary Fig. 4d-f).”

We obtained an IC₅₀ value of 1.8 μM for the positive control in our assay against our target ZIKV NS2B-NS3 protease. This is comparable to the reported IC₅₀ value of ~1 μM for the same compound (named as compound A in Kuiper *et al.*, Biochem Biophys Res Commun, 2017). The reported 18 nM inhibition of the positive control is targeting to Dengue2 NS2B-NS3 (named as compound 83 in Behnam *et al.*, J Med Chem, 2015). We assume that such significant difference in potency may be related to the sequence divergence in NS2B, as well as the structural dynamics between the NS2B co-factor and NS3 protease (open and closed conformations). Further studies are needed to better understand the observed IC₅₀ disparity across flaviviral NS2B-NS3 proteases. We have added the following in revised manuscript (Page 11, paragraph 1, line 372) to address this comment:

“Notably, the positive control compound showed an IC_{50} of 1.8 μ M in our assay (Supplementary Fig. 4g), consistent with a previous study (Biochem Biophys Res Commun, 2017). However, its potency against DENV2 NS3B-NS3 is significantly better, with a reported IC_{50} value in low nanomolar range. This observation underscores the challenge of developing broad-spectrum inhibitors targeting flaviviral NS2B-NS3 proteases, and highlights the importance of employing parallel biochemical and biophysical assays across a panel of NS2B-NS3 proteases.”

In our study, we used GCI based waveRAPID technology to measure the binding affinity. We clarified this method in the manuscript (line 361) by adding:

“waveRAPID is developed for compound and fragment screening and has demonstrated the capability to detect binding affinities in low millimolar range (EA FitzGerald et al., RSC Med. Chem., 2024).”

We obtained reliable results of positive control compound (Supplementary Fig. 4c), demonstrating the success of experimental setup and confirming the capability of this method to detect on-scale affinity binders for this protease. The unreliable results of fragments are likely due to the fragment-protein interactions being too weak to be detected. This is also supported by our biochemical assay, in which none of the fragment binders exhibited inhibitory activity.

Given the irreproducible data and limited information presented in Figure 6, we have moved it to Supplementary Fig. 4. We agree that other biophysical assays could be applied to test binding affinity. However, the fragment binders displayed very weak binding signals in our assay and did not exhibit inhibitory activity, suggesting that further characterization using alternative methods is unlikely to provide meaningful insights at this stage. Instead, the structural details on fragment-protein interactions are more informative for guiding fragment progression.

2. This paper shows a number of linked fragments, but not a single fragment has been synthesized or tested. This gives the paper an incomplete feeling compared to other papers in this journal.

Response: A new crystallographic structure of small molecule ASAP-0022538 binding to the active site has been added to Fig. 6. This small molecule recapitulated the poses and interactions of fragments **x0089**, **x1098** and **x0404**, and as such is an example of merged compound.

3. I have serious concerns about the nomenclature used in figure 2A. Line 161 notes reference 6 but uses standard Scheter and Berger nomenclature, which is a 50-year standard and the field, and should be appropriately reference. The colored patches in Fig 2A do not correspond to the Schechter and Berger subsites. Thus, the nomenclature is very confusing. This paper should use Schechter and Berger subsites. If they are not referring to those, then the paper should not use the standard nomenclature (S2, S3, S1 S1') as it is misleading.

Response: We corrected the reference using standard Schechter–Berger nomenclature (Berger A.; Schechter I. Mapping the active site of papain with the aid of peptide substrates and inhibitors. Philos. Trans. R. Soc. London B. Biol. Sci. 1970, 257 (813), 249–64). Fig. 2a has been modified accordingly.

4. The activity of cZiPro relative to bZiPro and gZiPro should be reported.

Response: We have addressed this comment by adding Fig. 1c and Supplementary Fig. 1 to compare the activity of constructs cZiPro, bZiPro and gZiPro. The following descriptions have been added in main text (Page 4, Paragraph 3, line149):

“The structural similarity between cZiPro and bZiPro supports their observed functional equivalence, as both enzymes displayed comparable enzymatic activities with K_m values around 6 μM and higher catalytic efficiencies than gZiPro (Fig. 1c), consistent with previous findings¹⁹. Moreover, all three proteases were similarly inhibited by bovine pancreatic trypsin inhibitor (BPTI), with IC_{50} values near 150 nM (Supplementary Fig. 1a). Notably, bZiPro and cZiPro exhibited similar thermal stability, with melting temperatures (T_m) of 47.3 °C and 46.6 °C, respectively, as measured by Differential Scanning Fluorimetry (DSF). In contrast, gZiPro was the most thermally stable, with a T_m of 50.8 °C, likely because its covalent linkage between the NS3 and NS2B cofactor (Supplementary Fig. 1b).”

Response to reviewers' comments

We appreciate the reviewers' time and effort in helping us improve our manuscript. We have considered Reviewer 2's feedback and revised the manuscript accordingly. Here is a point-by-point response to reviewer's comments.

REVIEWERS' COMMENTS

Reviewer #2 (Remarks to the Author):

While the authors addressed most of my previous comments, my enthusiasm for this study remains low since there is still no experimental validation of any of the predicted mergers, such as Z2451096209 and PV-004740099668.

Major comments

1. There remains a lack of an inhibitor developed using the information from the fragment screen in the present study. Although ASAP-0022538 recapitulates some of the fragment-protein interaction, it was identified in a previous study (ref #36) rather than constructed using the fragments from the present study. The impact of this study is limited without experimental evidence that the fragment-based screen can lead to new inhibitors.

Response: The work reported by Kenton et al (ref#36) is a follow-up study based on our fragment screening. **ASAP-0022538** is developed using our screened fragments as input for machine learning.

To address this comment, we have clarified in the main text as following:

*"To further analyse whether potency can be achieved with combined fragments, we assessed the binding pose of **ASAP-0022538** (PDB ID: 7I9O), a hit reported in a fragment-to-lead work using our screened fragments by Kenton et al (ref#36)."* (Page 9, Line 338)

Adding the following in main text (Page 9, Line 352):

"However, the utility of our fragments as the basis of inhibitor discovery was demonstrated by Kenton et al³⁶ who used our fragment data to initiate and complete a full fragment-to-lead campaign. The authors reported the use of machine learning of pharmacophoric motifs combined with iterative DMTA cycles in their campaign, progressing fragments to low-nanomolar potency³⁶."

Sentence (Page 9, Line 360): *"Merging algorithms that fully exploit chemical space and integrate key fragment-protein interactions are promising for identifying ASAP-0022538-like effective merges directly from the fragment screen, highlighting the necessity of algorithm development for fragment progression."* has been edit as:

*"Merging algorithms that fully exploit chemical space, integrate key fragment-protein interactions and reliably rank compounds from possible merges are promising for identifying ASAP-0022538-like effective merges directly from the fragment screen. Such an approach is described in Wills et al³⁷, and while that requires further refinement of the methodology, compound **ASAP-0022538** demonstrates that it is feasible."*

2. Relatedly, the phrase “development of Zika virus NS2B-NS3 protease inhibitors” in the title is misleading. First, there is only one compound in this study showing inhibition activity (i.e. ASAP-0022538). Second, ASAP-0022538 was identified in a previous study using machine learning (ref #36), rather than developed in this study.

Response: We thank the reviewer for highlighting that the title does not do the work justice. We have now updated it, to read:

“Combining a crystallographic fragment screen with deep mutational scanning provides a robust basis for discovery of inhibitors of Zika virus NS2B-NS3 protease”.

Minor comments

1. Line 417: should “ASAP-0022538 like” be “ASAP-0022538-like”?

Response: “ASAP-0022538 like” is corrected to “ASAP-0022538-like”

2. ASAP-0022538 is shown as an example of “active merge” in Figure 6C. However, the definition of “active merge” is unclear here.

Response: Indeed, including the term “active merge” was an oversight, we replaced ‘active merge’ in the main text as following:

“A crystal structure of ASAP-0022538 bound to ZIKV NS2B-NS3 (PDB ID: 7I9O) is shown as an example of merge design that achieves on-scale activity (low double-digit micromolar IC50).”